# Development of Effective Therapeutic Molecule from Natural Sources against Coronavirus Protease

**DOI:** 10.3390/ijms22179431

**Published:** 2021-08-30

**Authors:** Adewale Oluwaseun Fadaka, Nicole Remaliah Samantha Sibuyi, Darius Riziki Martin, Ashwil Klein, Abram Madiehe, Mervin Meyer

**Affiliations:** 1Department of Science and Innovation/Mintek Nanotechnology Innovation Centre, Biolabels Node, Department of Biotechnology, Faculty of Natural Sciences, University of the Western Cape, Bellville, Cape Town 7535, South Africa; nsibuyi@uwc.ac.za (N.R.S.S.); DRMartin@uwc.ac.za (D.R.M.); amadiehe@uwc.ac.za (A.M.); memeyer@uwc.ac.za (M.M.); 2Plant Omics Laboratory, Department of Biotechnology, Faculty of Natural Sciences, University of the Western Cape, Private Bag X17, Bellville, Cape Town 7535, South Africa; Aklein@uwc.ac.za; 3Nanobiotechnology Research Group, Department of Biotechnology, Faculty of Natural Sciences, University of the Western Cape, Bellville, Cape Town 7535, South Africa

**Keywords:** flavonols, SARS-CoV-2, M^pro^, natural products, MDs, MM/GBSA, docking, COVID-19, in silico approach, protein-ligand interaction, antiviral agents

## Abstract

The SARS-CoV-2 main protease (M^pro^) is one of the molecular targets for drug design. Effective vaccines have been identified as a long-term solution but the rate at which they are being administered is slow in several countries, and mutations of SARS-CoV-2 could render them less effective. Moreover, remdesivir seems to work only with some types of COVID-19 patients. Hence, the continuous investigation of new treatments for this disease is pivotal. This study investigated the inhibitory role of natural products against SARS-CoV-2 M^pro^ as repurposable agents in the treatment of coronavirus disease 2019 (COVID-19). Through in silico approach, selected flavonoids were docked into the active site of M^pro^. The free energies of the ligands complexed with M^pro^ were computationally estimated using the molecular mechanics-generalized Born surface area (MM/GBSA) method. In addition, the inhibition process of SARS-CoV-2 Mpro with these ligands was simulated at 100 ns in order to uncover the dynamic behavior and complex stability. The docking results showed that the selected flavonoids exhibited good poses in the binding domain of M^pro^. The amino acid residues involved in the binding of the selected ligands correlated well with the residues involved with the mechanism-based inhibitor (N3) and the docking score of Quercetin-3-O-Neohesperidoside (−16.8 Kcal/mol) ranked efficiently with this inhibitor (−16.5 Kcal/mol). In addition, single-structure MM/GBSA rescoring method showed that Quercetin-3-O-Neohesperidoside (−87.60 Kcal/mol) is more energetically favored than N3 (−80.88 Kcal/mol) and other ligands (Myricetin 3-Rutinoside (−87.50 Kcal/mol), Quercetin 3-Rhamnoside (−80.17 Kcal/mol), Rutin (−58.98 Kcal/mol), and Myricitrin (−49.22 Kcal/mol). The molecular dynamics simulation (MDs) pinpointed the stability of these complexes over the course of 100 ns with reduced RMSD and RMSF. Based on the docking results and energy calculation, together with the RMSD of 1.98 ± 0.19 Å and RMSF of 1.00 ± 0.51 Å, Quercetin-3-O-Neohesperidoside is a better inhibitor of M^pro^ compared to N3 and other selected ligands and can be repurposed as a drug candidate for the treatment of COVID-19. In addition, this study demonstrated that in silico docking, free energy calculations, and MDs, respectively, are applicable to estimating the interaction, energetics, and dynamic behavior of molecular targets by natural products and can be used to direct the development of novel target function modulators.

## 1. Introduction

Viral infection is one of the major challenges faced by human health, and many viral diseases are correlated with high morbidity and mortality rates in humans. Previously, viral diseases such as influenza, dengue, HIV, and coronaviruses have resulted in epidemics or global pandemics, claiming many lives. Currently, the world is battling with coronavirus diseases 2019 (COVID-19) caused by severe acute respiratory syndrome coronavirus 2 (SARS-CoV-2) which has resulted in approximately 4 million deaths as of July 2021. The emergence of variants of this disease has also been challenging to the developed vaccines, and, as it stands, there are no effective therapeutic interventions against this disease. Proteins associated with viral infection can serve as molecular targets for disease prevention and treatment. Molecules involved in viral DNA replication and protein synthesis can be pivotal to these processes. Targets such as Papain-like protease (PLpro), 3-chymotrypsin like protease (3CLPro) also called main protease (M^pro^), RNA-dependent RNA polymerase (RdRp), and helicase have been reported as potential points of treatment development for SARS-CoV-2 infection [1,2,3]. The SARS-CoV-2 spike protein and angiotensin-converting enzyme-2 (ACE-2) were also identified as promising targets for disease prevention [4,5,6]. The non-structural proteins essential for the replication of viral particles are specifically generated by PLpro and M^pro^, defining their roles in viral replication and outlining their inhibition as potential anti-SARS-CoV-2 treatments [7,8,9].

Biologically active molecules have been screened against M^pro^ as either repurposed drugs or in the process of lead identification. Muhammad et al. [10] screened a library of phytochemicals against M^pro^ and revealed the potential usage of molecules from natural sources as anti-COVID-19 druggable candidates. In another study, bioactive medicinal plants were assayed against M^pro^ using in silico docking and pharmacological screening. Selected molecules from alkaloids and terpenoids were identified as inhibitors of this target, with a highly conserved inhibitory pattern to both SARS-CoV-2 and SARS-CoV [11]. Moreover, nucleopeptides and Opuntia-derived phytochemicals were suggested as M^pro^ potential inhibitors [12,13]. Therefore, the inhibition of protein activity of M^pro^ may potentially suppress coronavirus transmission.

Natural products of low molecular weight from plant sources are potent therapeutic agents for many diseases and some of these agents possess antiviral properties. While many natural products are fundamentally utilized as crude extracts, the purification of their active ingredients is essential for the prediction of their properties associated with the pharmacokinetics and pharmacodynamics of a drug molecule.

Accumulating evidence suggests that plant chemicals, for example, polyphenols and their functional derivatives such as flavonoids, saponins, and lignans can alter cellular functions, membrane permeability, and viral replication [14]. The role of phytochemicals has also been implicated in cell migration and proliferation, metabolism regulation (phytosterol, flavanols, anthocyanidins, cinnamic acids, etc.) [15,16], inflammatory processes (quercetin, kaempferol, etc.) [17], redox modulation (phenolics, curcumin, resveratrol, etc.) [18,19], and angiogenesis (astaxanthin) [20]. Ethanolic extract of *Ficus benjamina* has been shown to contain some active compounds such as rutin, kaempferol 3-O-rutinoside, and kaempferol 3-O-robinobioside, which were effective against herpes simplex [21]. Equally, the phytochemicals homolycorine and 2-O-acetyllycorine isolated from *Leucojum vernum* were shown to be effective against HIV-1 [22]. Rutin is a glycosylated flavonoid with a 3-rutinoside substitution. Its antiviral activity has been studied against avian influenza virus [23], herpes simplex [21], and parainfluenza-3 virus [24].

The world population is largely dependent on therapies from plant origin [25]. Compounds isolated from this source have little or no side effects with high biological specificity, chemical diversity, and targets multiple host sites by diverse pathways with negligible cost [26]. It was previously demonstrated that flavonoids, among other active compounds from plant sources, have been used for the treatment of HIV, herpes simplex, and influenza viruses, due to their antiviral properties [27]. These compounds inhibit viral replication, proteases, and reverse transcription [28]. Compounds such as quercetin, rutin, and myricetin have also been identified with similar properties [29].

In this study, five glycosylated flavonoids, belonging to the class of Flavonol, were selected to evaluate the inhibitory process of SARS-CoV-2 M^pro^. These compounds include Quercetin 3-Rhamnoside, Quercetin-3-O-Neohesperidoside Myricetin 3-Rutinoside, Rutin, and Myricitrin. The mode of antiviral action of Quercetin 3-Rhamnoside, a phytochemical derived from *Houttuynia cordata* (Aerial parts), is by targeting influenza A/WS/33 virus through the inhibition of replication in the early stage of viral infection by indirect interaction with virus particles [30]. The major active compound of *Trollius ledebouri Reichb*., a popular medicinal plant in China, known to possess antibacterial and antiviral activities used in the treatment of acute myringitis, cold, sore throat, and fever, is Quercetin-3-O-Neohesperidoside [31,32]. Pharmacological Screening suggests that this compound possesses activity against inflammation [33]. Myricetin and its derivatives derived from Myricaceae are known for their nutraceuticals value with activities including anticancer, antioxidant, anti-inflammatory, and antidiabetic [34]. Flavonoids, such as Myricitrin, Quercetin, and Kaempferol, are structurally related to Myricetin [35]. Myricetin differs from Myricitrin in that the hydrogen atom of Myricetin at carbon 3 of the C-ring is substituted with rhamnose. At this same position, rutinoside can be substituted to form Myricetin 3-Rutinoside. Properties of Myricetin against bacteria and viruses were established by Cai, et al. [36]. Recently, the antiviral property of Myricetin and Myricitrin was demonstrated against swine fever virus protease. In the study, 65 flavonoids were screened against the swine fever viral protease using the fluorescence resonance energy transfer method. Myricetin and Myricitrin showed inhibitory potential against the target, suggesting their role in the development of anti-swine fever virus agents based on their scaffold [37]. Rutin is otherwise known as rutoside or quercetin-3-O-rutinoside. Structurally, this glycosylated flavonoid consists of rutinose linked to Quercetin. The neuroprotective properties of this flavonoid were reported against a cisplatin-induced neurotoxic rat model. Rutin was able to restore the expression levels of PON-1, PON-2, PON-3, PPAR-δ, and GPX to normal after being dysregulated by cisplatin [38]. In another study, the anticancer effect of rutin on breast cancer (BC) was evaluated by in silico and Western blot approaches. The study concluded that rutin targets c-Met kinase and in turn reduces tumor size and inhibits cell proliferation, migration, and invasion in BC [39].

The availability of the resolved structures of the essential protein components of this disease has improved the search for effective therapeutic molecules through in silico approaches. Molecular docking calculations have gained enormous attention in the search for novel bioactive molecules and other discovery projects. This approach is an alternative to experimental techniques due to its speed and accuracy to predict both the binding affinities and poses of ligands within molecular targets. The prediction accuracy of docking algorithms can suggest bioactive compounds with higher true positive rates which can, in turn, reduce experimental challenges [40]. Therefore, the identification of antiviral drugs to combat infectious diseases can be dependent on the investigation of available compounds from natural origin with antiviral properties [41]. With limited therapeutic options for the treatment of COVID-19, five antiviral glycosylated flavonoids were screened against one of the most essential components of SARS-CoV-2 to investigate their inhibitory role against the SARS-CoV-2 main protease.

## 2. Results

Lead discovery strategies consist of screening low molecular weight compounds (ligands) against biological targets with clinical relevance. Molecular docking is an essential approach for drug design, most especially in the identification of lead or hit agents. In this study, the inhibitory role of five natural products (glycosylated flavonoids) was queried against M^pro^ in order to ascertain their therapeutic properties against COVID-19. As it stands, there is no 100% safe vaccine, and effective therapeutic agents are far from being discovered against this deadly disease.

### 2.1. ADME/Tox Prediction

The pharmacokinetics and toxicological properties of the ligands (Table 1 and Table 2) were analyzed according to previous methods to investigate how molecules can access the target site of M^pro^ after entering the bloodstream. This analysis is also crucial for analyzing the efficacy of molecules [42,43]. All parameters were within the ROF cut-off range for the test compounds and present no bystander toxicity effects since toxicity is the main task in developing new medications. Ames toxicity, carcinogenic properties, and rat acute toxicity were predicted in the current investigation.

### 2.2. Docking Calculations

The protein-ligand interactions for all the complexes after the docking procedure were produced by Proteins Plus at https://proteins.plus/ (accessed on 1 August 2021) as depicted in Figure 1. The model representation of the best pose against decoy poses was also presented using M^pro^-Quercetin-3-O-Neohesperidoside (Figure 2). The binding properties, such as the scoring functions of Autodock Vina and MM/GBSA, the number of hydrogen bond integrations, and types of residues and their distances (Å) are tabulated in Table 3. In addition, the possible residue interaction crucial to the inhibition process of M^pro^ by these flavonoids was proposed in Figure 3.

### 2.3. Molecular Dynamic Simulation

To associate structural and mechanistic information with experimental data, the MDs were carried out. The M^pro^-ligand complexes were computationally simulated for 100 ns to decipher the complex stability and dynamic behavior as presented in Figure 4, Figure 5 and Figure 6, and Table 2. The root-mean-square deviation (RMSD) of the five ligands were plotted against 1000 frame indexes for 100 ns (Figure 4a). The root-mean-square fluctuation (RMSF) of the C-alpha of the protein complexed with these ligands was also plotted against residues (Figure 4b). The ligand properties were also taken into account by plotting the radius of gyration (rGyr) for all five complexes against the frame index over the 100 ns simulation time (Figure 5). The simulation properties were calculated as the mean ± SD for RMSD, RMSF, and rGyr (Table 4). Finally, protein interactions with the ligands were monitored throughout the simulation. These interactions were categorized by type and summarized as shown in Figure 6.

## 3. Discussion

Biologically active molecules from natural sources have been explored for their use in the treatment of diseases. The relationship between secondary metabolites and disease prevention has been a focal point in medicine. Growing knowledge of disease pathogenesis and molecular targets by small molecules from natural products may provide novel strategies to prevent or manage various diseases. With various challenges associated with various discovered vaccines, coupled with limited therapeutic options against COVID-19, in silico approaches may be employed to source effective compounds against SARS-CoV-2 macromolecules. Compounds with antiviral properties have been isolated and purified from plant origin [44], some of which possess numerous advantages that assist in inhibiting viral replication, viral penetration in the host cell [45]. Docking approaches primarily identify correct ligand orientations and accurately predict binding affinities.

### 3.1. Drug-Likeness (ADME/Tox) Analysis

Lipinski’s ROF was designed to set guidelines for the drugability of compounds. The efficacy of drug candidate does not directly correlate with Lipinski’s rule of five for disease treatment and meeting these criteria does not ascertain drug-likeness [46]. However, challenges associated with oral activities may arise for Lipinski’s violation [47]. The Lipinski’s parameters are closely associated with oral bioavailability, intestinal permeability and acceptable aqueous solubility [42]. The selected glycosylated flavonoids are in agreement with the oral drugability and bioavailability of Lipinski’s rule of five criteria. Veber’s rule suggests that compounds with hydrogen bond acceptors and donors ≤ 12 have better oral bioavailability [48]. Quercetin-3-O-Neohesperidoside, Quercetin 3-Rhamnoside, Myricitrin, Rutin, and Myricetin 3-Rutinoside were all predicted to meet the majority of the criteria for an orally active drug molecule.

### 3.2. Docking Analysis

The docking program Autodock Vina was used to produce different poses of the ligands and further rank these ligands based on their scoring function. This software was able to pinpoint the native docking pose among the decoy poses and further predicted the binding affinities of the ligands based on the scoring power. M^pro^-N3 was ranked second highest based on the negative binding energy of −16.5 Kcal/mol, while Myricitrin was ranked lowest with −9.1 Kcal/mol. The intermolecular interaction of Quercetin-3-O-Neohesperidoside as a model was ranked first with a binding energy of −16.8 Kcal/mol. A similar study reported lower negative energy (−8 kcal/mol) for Quercetin-3-O-Neohesperidoside as opposed to −16.8 Kcal/mol obtained in this study [49]. The reason for this could be attributed to the method of preparation and minimization of both the ligand and the receptor, and also the size of the generated grid box before docking. In actual fact, the bigger the grid box created prior to docking, the more time-consuming the simulation process which could, in turn, affect the output in terms of binding score.

The scoring function of docking calculation provides a quick but crude estimation of the binding affinity between the ligand and target [50]. During docking calculations, scoring functions are used to extrapolate the docking poses derived by docking algorithms. These functions facilitate the binding affinities to attain high throughput screening. It was previously reported that docking calculations often fail to evaluate binding energy when compared to experimental values [51]. Therefore, the Prime MM/GBSA software was used to re-score the ligand binding affinities due to its rapid force-field-based method used for computations [52]. Comparing the dock scores and MM/GBSA rescoring methods, MM/GBSA provides more accurate results. The MM/GBSA rescoring ranked the binding affinities in the order: Quercetin 3-O-Neohesperidoside > Myricetin 3-Rutinoside > Quercetin 3-Rhamnoside > Rutin > Myricitrin, to corroborate the docking scores. Although natural products are sometimes underestimated when compared to Western medicines, drugs such as paclitaxel aspirin, digitalis, and reserpine with therapeutic efficacy were derived from natural products [53]. With many advantages associated with natural products the discoveries of therapeutic compounds deserve the efficacy of natural sources.

Judging from the number of hydrogen bond interactions exhibited by the ligands, CYS145, GLU166, HIS41, THR190, and ASN142, play a crucial role in the M^pro^ binding process. The most notable interactions involve the carboxyl and amino groups of charged residues (GLU166, HIS41, and HIS164) and polar residues (ASN142 and CYS145). These residues provide rich hydrogen bond donors or acceptors between the amino or carboxyl groups of M^pro^ and the poly-hydroxyl groups of the ring structures of the ligands. In addition to the hydrogen bond interactions, Quercetin 3-O-Neohesperidoside and Myricitrin formed a π-π stacking between the dihdroxylphenyl D aromatic ring and HIS41 residue of M^pro^ with a distance of 1.49 and 5.37 Å, respectively. Pi-pi (π-π) stacking and π-cation interactions are crucial for favorable electron correlation and electrostatic contributions, respectively [54]. The difference in bond length could also contribute to the reason why Quercetin 3-O-Neohesperidoside has a higher negative energy compared to Myricitrin. Protein-ligand interactions are fundamental to biological processes. Ligand interactions cause conformational changes in both the ligand and protein, which further induces a series of events resulting in diverse cellular activities [55]. Jin et al. [56], who discovered the crystal structure of M^pro^ complexed with N3 using computer-aided software, further analyzed the M^pro^ residues involved in binding interactions. Amino acids such as PHE140, GLN189, GLY143, GLU166, THR190, CYS145, and HIS163 were reported to be crucial in the inhibition process of M^pro^ through hydrogen bonding. In addition, several hydrophobic interactions were also reported.

The possible mechanism of inhibition of SARS-CoV-2 was proposed through residue linkage and different interactions with poly hydroxyl and oxygen atoms of natural products using a flavonoid (Quercetin 3-O-Neohesperidoside) as a representative. Although a number of studies have employed in silico approaches to uncover the interactions between different receptors and ligands, and have also used similar compounds for different targets, none of these studies have been able to elucidate or propose the inhibitory mechanism by which flavonoids and their derivatives interact with their respective targets.

This structure is composed of five aromatic rings (A–E). The parent compound flavonol (3,4-dihydroxyphenyl-5,7-dihydroxychromen-4-one), consisting of poly hydroxyl groups at positions A-7 and 5, and B-2 and 4, has the ability to form hydrogen bonds with THR190 and GLU166, respectively. The hydroxyl atoms of Oxyoxane and 6-methyloxane were also capable of hydrogen linkage with residues such as GLN189, HIS164, CYS145, and GLY143. π-π stacking was also found in the aromatic B-ring of the parent compound and HIS41. Other hydrophobic interactions include MET165, ARG188, ASP188, and HIS41. These multiple interactions may suggest a good binding affinity of flavonoids and other natural products with targets and further inhibition of their activities.

Gyebi et al. [11] identified the importance of CYS145 and HIS41 as conserved catalytic dyad residues of SARS-CoV-2 M^pro^. Interestingly, all of the screened molecules in this study interacted with one of the two amino acids. The atoms of Quercetin 3-O-Neohesperidoside and Myricitrin interacted with these residues, suggesting their role as candidates or leads for drug development against this disease. Natural products with biological activity, with reference to past events, may be good sources for the synthesis of drugs with antiviral properties.

### 3.3. Molecular Dynamic Simulation

MDs were carried out to generate dynamic representation at atomic spatial resolution for M^pro^-ligand complexes for all the studied compounds, and to establish a more reliable mechanism for illustrating protein interactions with small molecules. After the molecular docking of the selected glycosylated flavonoids, MDs of the complexes were evaluated for stability in terms of residue flexibility and bond types over the course of 100 ns. The stability and structural flexibility of these complexes were studied through RMSD and RMSF plots, respectively.

The RMSD is a measure of atomic distances or coordinates, such as the C-alpha protein backbone. The MD trajectory analysis of the complexes showed that Quercetin 3-Rhamnoside, Myricitrin, and Myricetin 3-Rutinoside are more stable than Rutin over 100 ns of simulation. The obtained RMSD values ranged from 1.81 ± 0.30 to 3.05 ± 0.57 Å. Rutin had the lowest RMSD (1.81 ± 0.30 Å) followed by Quercetin-3-O-Neohesperidoside with RMSD of 1.98 ± 0.19 Å. The lower the RMSD the less deviation from the backbone and the greater the stability [57]. The stability of these compounds is in the order of Rutin > Quercetin-3-O-Neohesperidoside > Quercetin 3-Rhamnoside > Myricetin 3-Rutinoside > Myricitrin. The slight variations observed may be attributed to complex conformations [58]. The average deviation of the complexes was monitored by the RMSF from a reference point over 100 ns. To probe how the molecules affect the dynamics of the backbone atoms of M^pro^, the RMSF values were calculated. The RMSFs of the complexes were ≤1.31 Å. Quercetin-3-O-Neohesperidoside had the lowest RMSF of 1.00 ± 0.51 Å followed by Rutin with an RMSF of 1.15 ± 0.58 Å. Quercetin 3-Rhamnoside, Myricitrin, and Myricetin 3-Rutinoside had the highest RMSF of 1.31 ± 0.55, 1.27 ± 0.58, and 1.25 ± 0.65 Å, respectively.

Besides the C and N-terminal RMSFs that are expected to fluctuate maximally, fluctuations were observed around the residues 50 and 180. These regions could be loop regions with additional conformational flexibilities. The RMSF values of the systems relative to each amino acid residue of the complexes were evaluated to compare the residues’ flexibility. The RMSF reported the flexible region of the complexes and displayed that none of the important catalytic residues present in the active site region has shown an RMSF value of more than 1.31 Å. This serves to confirm that the catalytic machinery present in the M^pro^ active site was not distorted upon binding of the compounds. The amino acids, namely, THR26, HIS41, CYS44, SER47, MET49, TYR54, ASN142, GLY143, SER144, CYS145, HIS164, MET165, GLU166, LEU167, ASP187, THR190, GLN192, ARG217, THR304, and PHE305 have displayed flexibility upon the binding of the studied compounds. Regardless of the nature of these residues within the active site of M^pro^, they were relatively stable in complex systems indicating the binding nature of the compounds. With respect to all these complexes, the residues fluctuate with respect to the nature of the compound in the active site and types of interacting atoms. Particularly, THR190, GLU16, HIS163, PHE140, GLY143, CYS145, and GLN189 fluctuated more compared to other amino acids in the active site indicating their flexible nature and therefore could be a good determinant in the identification of inhibitory molecules. 

The rGyr was used to demonstrates the size and the compactness of the complexes [59] relative to the RMSD of the protein. Rutin has the lowest average rGyr of 4.57 ± 0.05 Å while Myricetin 3-Rutinoside has the highest average value of 6.09 ± 0.09 Å. Although the values of the rGyr for each of the complexes were high, there was no visible fluctuation throughout the simulation process. According to a study by Cherrak et al. [49], who observed a low rGyr of 3 natural compounds in the range of 2.2–2.3 nm and did not induce structural changes in the protein, a high rGyr value in this study could be associated with protein conformational changes which could alter the normal function of the SARS-CoV-2 M^pro^.

In tracking the dynamical changes following protein modification, the possible interactions such as hydrogen bond, hydrophobic interaction, water bridges, and ionic interaction were studied for the five complexes simulated over a 100 ns period. GLU166 interacted almost 90% of the entire simulation period between Quercetin-3-O-Neohesperidoside, Quercetin 3-Rhamnoside, Myricitrin, and Rutin and M^pro^ using both hydrogen bond and water bridges. To a lesser extent, Myricetin 3-Rutinoside interacted with this residue about 20% of the 100 ns. In addition, the number of observable hydrogen bonds of the ligands in the active site of Mpro is in the order of Quercetin 3-Rhamnoside < Rutin < Myricitrin < Myricetin 3-Rutinoside < Quercetin-3-O-Neohesperidoside. Hydrogen bonds, among other interactions, play a pivotal role in the stability of a protein’s secondary structure. Quercetin 3-O-Neohesperidoside was found to be stable in the active site of M^pro^ throughout the 100 ns simulation with an average RMSD of 1.98 ± 0.19 Å and. The slight variation of the RMSD observed over the simulation time could be attributed to the receptor undergoing conformational changes. In short, the docking result of M^pro^-Quercetin 3-O-Neohesperidoside complex through specific interactions and energy calculations showed that Quercetin 3-O-Neohesperidoside binding to M^pro^ is energetically favorable. In addition, the stability and structural flexibility of the complex may be an indication that Quercetin 3-O-Neohesperidoside may be repurposed as a potential inhibitor of SARS-CoV-2.

## 4. Materials and Methods

### 4.1. ADME/Tox Analysis (Drug-Likeness)

The Schrodinger QikProp module, SwissADME (http://www.swissadme.ch/index.php, accessed on 1 August 2021), and AdmetSAR (http://lmmd.ecust.edu.cn/admetsar1/predict, accessed on 1 August 2021) were combined and used to analyze the physicochemical properties, water solubility, pharmacokinetics, lipophilicity, and drug-likeness, of selected ligand. For toxicity, the Ames toxicity, carcinogenic properties, acute oral toxicity, and rat acute toxicity were predicted.

### 4.2. Protein Preparation

An in silico molecular docking simulation approach was employed in this study. Briefly, the SARS-CoV-2 main protease (M^pro^), with the PDB ID: 6LU7, was retrieved from the protein data bank (PDB) at https://www.rcsb.org/ (accessed on 1 August 2021). During structural inspection with PDBSum, M^pro^ consisted of 306 amino acids, 84 water molecules, and a co-crystalized ligand (N3). The M^pro^ resolution was 2.16 Å. The protein preparation was carried out by UCSF Chimera prior to the docking simulation, and the co-crystalized ligand (N3) was taken as the control. Water molecules, heteroatoms, and co-crystallized solvent were removed and polar hydrogens and partial charges were added to the structure protein structure during preparation. This was taken as the receptor.

### 4.3. Ligand Selection and Preparation

Five bioactive glycosylated flavonoids were identified based on the literature search and previously reported by Cherrak et al. [49] as potential Mpro inhibitors. These ligands were retrieved from PubChem at https://pubchem.ncbi.nlm.nih.gov/ (accessed on 1 August 2021) and individually saved in the sdf file format. Preparation such as energy minimization among other parameters was carried out to generate an accurate model that enumerates different structural and chemical possibilities of these ligands by Molegro Molecular Viewer (MMV). The outputs were individually saved in the mol2 file format prior to the docking calculation.

### 4.4. Molecular Docking Calculation

The interaction of the prepared ligands and N3 were investigated by docking them into the active site of the receptor (M^pro^) individually using the Autodock Vina software with built-in UCSC Chimera. The 2D interaction diagram, poses, and their energies were further computed.

### 4.5. ∆G Energy Calculation of the Docked Complexes

The ligand-binding energies (Δ*G*_bind_ in kcal/mol) were calculated from the docking result using the Prime technology (MM/GBSA) module from Schrodinger. The energies of the ligand–Mpro complex was estimated individually using the equation: Δ*G*_bind_ = *G*_complex_ − *G*_protein_ − *G*_ligand_ where Δ*G*_bind_ is the binding free energy and *G*_complex_, *G*_protein_, and *G*_ligand_ are the free energies of complex, protein, and ligand, respectively [60,61]. In this study, only the free binding energies of the ligands were shown.

### 4.6. Molecular Dynamics Simulation (MDs)

This process of MDs was carried out by the Desmond module of the Schrodinger suite. Briefly, a model system was built individually for the five complexes using the system builder module in maestro. Here, the model system was built to include M^pro^, ligands, explicit solvent, and counter ions (Na^+^ and Cl^−^). A value of 0.15 M is approximately the physiological concentration of monovalent ions. The solvent model was predefined as transferable intermolecular potential with 3 points (TIP3P) water molecules, box shape was set as orthorhombic, box volume was calculated and minimized accordingly, while the optimized potentials for liquid simulations (OPLS_2005) was used to prepare the force field. The temperature and pressure of the model system were maintained at 300° Kelvin and 1.01325 bar, respectively [62]. Finally, the simulation was carried out at 100 nanoseconds (ns). The output was analyzed by simulation interactions diagram module and MS-MD trajectory analysis.

## 5. Conclusions

The contribution of biologically active molecules from natural sources has been increasingly explored and exploited in the area of drug design using various techniques for target identification and function attribution. Therefore, key protein identification as a possible molecular target may by an interesting strategy for disease therapy and management. The current pandemic caused by SARS-CoV-2 has resulted in approximately 4 million mortalities with an increasing rate of infection globally. However, proteins such as Mpro, among other targets, has been identified as a potential therapeutic target. The M^pro^ of SARS-CoV-2 is involved in the processing of polyproteins that are translated from the viral RNA. Therefore, the inhibition of M^pro^ could serve as an attractive drug target to prevent disease transmission.

This study identified selected glycosylated flavonoids with reported biological activities as a potent inhibitor of SARS-CoV-2 M^pro^. Through an in silico procedure, Quercetin 3-Rhamnoside was identified as the most SARS-CoV-2 M^pro^ inhibitor. The molecular docking result showed that these flavonoids displayed strong inhibitory activity against M^pro^. Quercetin 3-Rhamnoside interacted better than the reported mechanism-based inhibitor (N3) of the SARS-CoV-2 Mpro. The MDs result reflected good dynamic behavior and structural stability of M^pro^ complexed with all the ligands. Since substituted flavonoids undergo reduced dietary modification in the gastrointestinal tract, this study hypothesized that these classes of natural products may be useful in drug design or repurposed drugs for disease therapy. In addition, target identification with bioactive molecules using both docking calculation and molecular simulation approaches may suggest a pivotal source of novel probe molecules against new and biologically relevant targets of deadly diseases such as cancer and SARS-CoV-2.

## Figures and Tables

**Figure 1 ijms-22-09431-f001:**
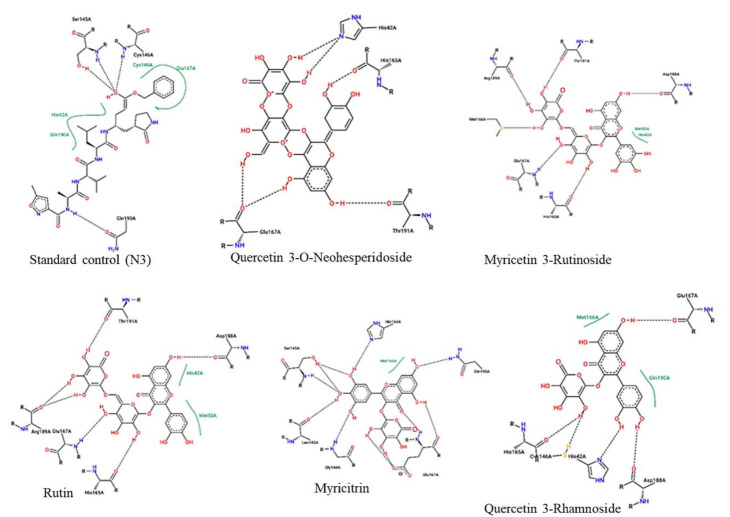
2-dimentional protein–ligand interactions created for docked ligands and N3 into the active site of M^pro^ ranked according to their binding energies. Black bond interactions showed H-bonds between the atoms of the ligands and the residues of the receptor.

**Figure 2 ijms-22-09431-f002:**
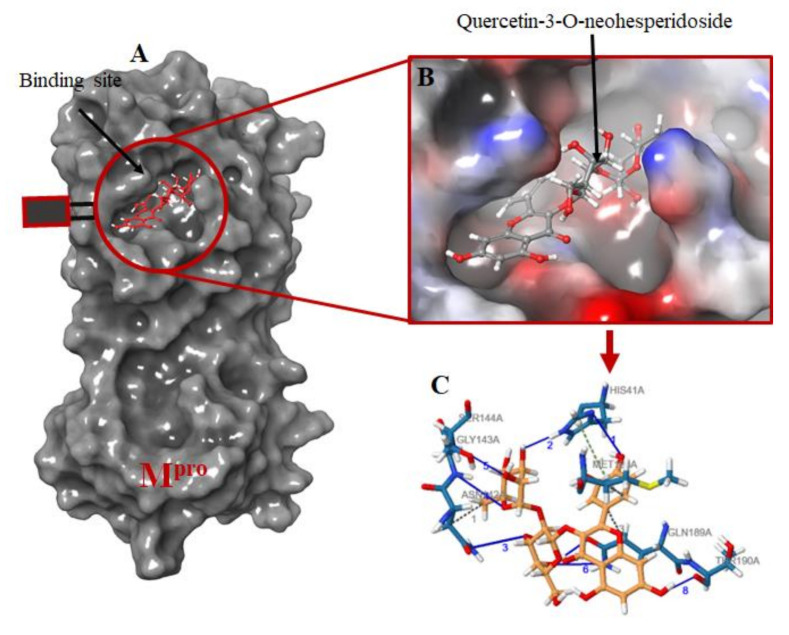
Molecular docking of the ligands into M^pro^ binding site. (**A**) the docking of the selected ligands in M^pro^ binding pocket; (**B**) Quercetin-3-O-Neohesperidoside seats perfectly in the Mpro binding pocket; (**C**) interacting atoms of Quercetin-3-O-Neohesperidoside and M^pro^ residues in the binding pocket. Note: blue lines indicate hydrogen bond interaction; green dotted lines indicate pi-stacking, while gray dotted lines depict hydrophobic interactions.

**Figure 3 ijms-22-09431-f003:**
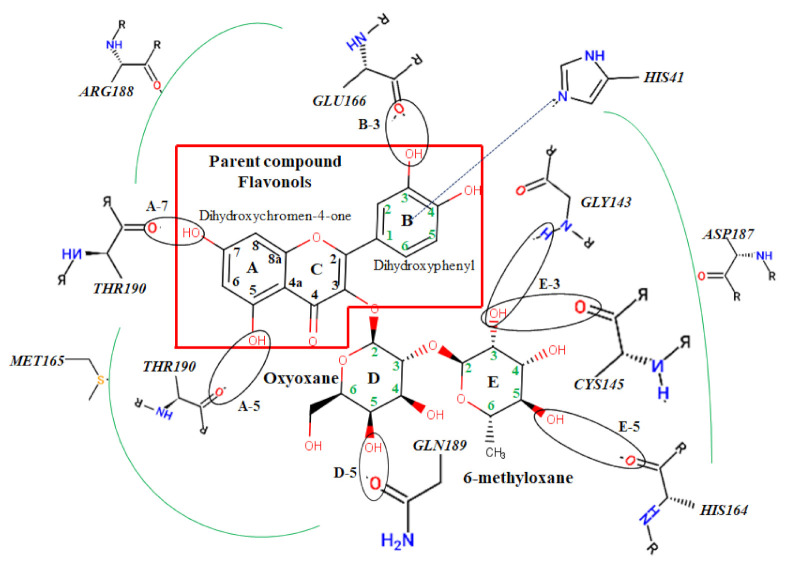
Atomic interaction of flavonoids with Mpro residues. While several residues form hydrophobic interactions with the ligands, other interactions such as hydrogen, π-cation, and π-π stacking were also involved in the inhibition mechanism of M^pro^.

**Figure 4 ijms-22-09431-f004:**
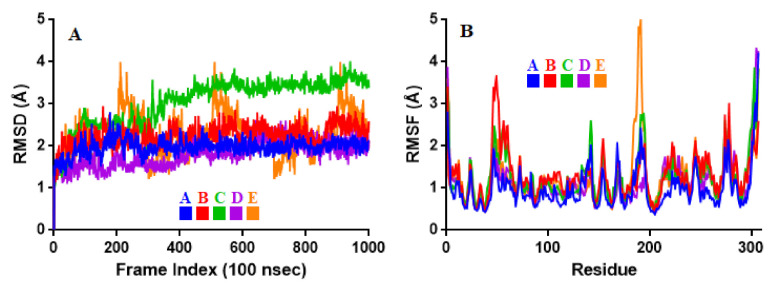
Stability and flexibility of the selected flavonoid derivatives complexed with M^pro^ over the course of 100 ns. (**A**) is the RMS deviation and (**B**) is the RMS functions of the Cα of each complexes. Color codes denote: Quercetin-3-O-Neohesperidoside (A), Quercetin 3-Rhamnoside (B), Myricitrin (C), Rutin (D), and Myricetin 3-Rutinoside (E).

**Figure 5 ijms-22-09431-f005:**
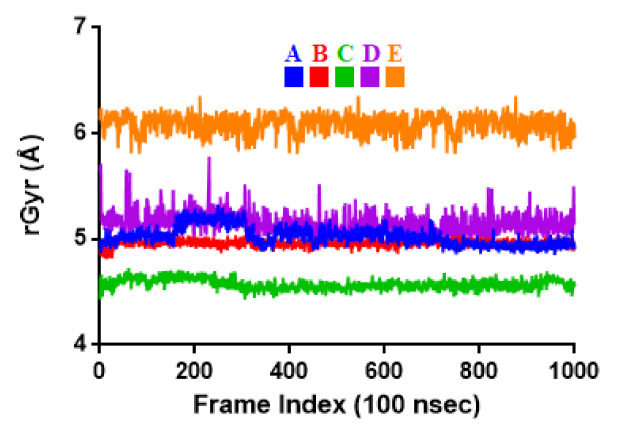
Radius of gyration (rGyr) plot for M^pro^ and the selected flavonoid derivatives within the simulation time of 100 ns. Color codes denote: Quercetin-3-O-Neohesperidoside (A), Quercetin 3-Rhamnoside (B), Myricitrin (C), Rutin (D), and Myricetin 3-Rutinoside (E).

**Figure 6 ijms-22-09431-f006:**
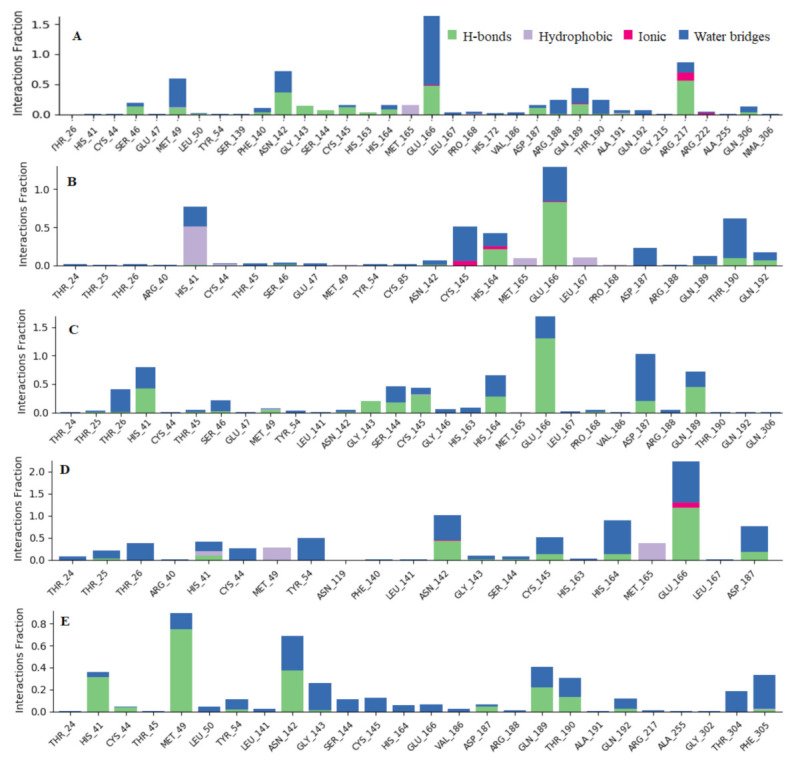
Observed M^pro^-ligands interaction during the 100 ns MD simulation. Interactions include; hydrogen bonds, hydrophobic, ionic and water bridges. Letter codes indicate: Quercetin-3-O-Neohesperidoside (**A**), Quercetin 3-Rhamnoside (**B**), Myricitrin (**C**), Rutin (**D**), and Myricetin 3-Rutinoside (**E**).

**Table 1 ijms-22-09431-t001:** Properties of the selected compounds.

Compound	ID ^a^	M.W ^b^	ROF ^c^	QplogHERG ^d^	QplogPoW ^e^	QplogKP ^f^	Donor HB	Acceptor HB	QplogS ^g^	QplogBB ^h^
Quercetin 3-O-Neohesperidoside	5748416	610.5	2	−6.449	−1.998	−6.423	9	20.55	−2.932	−4.728
Myricetin 3-Rutinoside	44259428	626.5	2	−6.394	−2.455	−5.583	10	21.3	−2.341	−4.306
Quercetin 3-Rhamnoside	5353915	448.3	2	−5.451	−0.55	−6.101	6	12.05	−3.196	−3.312
Rutin	5280805	610.5	2	−5.238	−2.495	−7.251	9	20.55	−2.175	−4.503
Myricitrin	5281673	464.3	2	−5.463	−1.045	−6.589	7	12.8	−2.779	−3.48

^a^ Compound ID from PubChem database. ^b^ Formula weight of the compounds (acceptable range: 130.0–725 g/mol). ^c^ Number of permissible violations of Lipinski’s rule of five (acceptable range: maximum is 4). ^d^ Predicted IC_50_ value for blockage of HERG K^+^ channels (concern below −5.0). ^e^ Predicted octanol/water partition coefficient log p (acceptable range: −2.0 to 6.5). ^f^ Predicted skin permeability, log Kp (acceptable range: −8.0 to −1.0). ^g^ Predicted aqueous solubility; S in mol/L (acceptable range: −6.5 to 0.5). ^h^ Predicted brain/blood partition coefficient (acceptable range: −3.0 to 1.2). Donor HB (≤10); Acceptor HB (≤5).

**Table 2 ijms-22-09431-t002:** Toxicity analysis of the selected compounds predicted by AdmetSAR.

Compound	ID	Ames Toxicity	Carcinogens	Acute Oral Toxicity	Rat Acute Toxicity
Quercetin 3-O-Neohesperidoside	5748416	AT	NC	III	2.2619
Myricetin 3-Rutinoside	44259428	NAT	NC	III	2.4984
Quercetin 3-Rhamnoside	5353915	NAT	NC	III	2.5458
Rutin	5280805	NAT	NC	III	2.4984
Myricitrin	5281673	NAT	NC	III	2.5458

Note: AT: Ames toxic NAT: Non Ames toxic; NC: Non-carcinogenic; Category-III means (500 mg/kg > LD50 < 5000 mg/kg).

**Table 3 ijms-22-09431-t003:** Binding energies of flavonoids docked against M^pro^.

Name	Dock Score	∆G Bind	H-Bond	Residues (Å)	Other Bond (Å)
Standard	−16.5	−80.88	6	CYS145 (2.13), LEU141 (2.76), PHE140 (2.02), GLU166 (1.68,1.83, 2.05)	Salt bridges (2)
Quercetin 3-O-Neohesperidoside	−16.8	−87.60	5	GLY143 (2.76), CYS145 (2.11), GLN189 (2.11), THR190 (1.76), HIS41 (2.30)	π-π stacking HIS41 (1.49)
Myricetin 3-Rutinoside	−12.9	−87.50	7	CYS145 (2.08), ASN142 (1.75), GLU166 (1.98), THR190 (2.21), ARG188 (1.97), HIS164 (1.90,1.98)	
Quercetin 3-Rhamnoside	−10.3	−80.17	4	LEU141 (1.49), THR190 (1.78), GLU166 (2.01), HIS164 (1.81)	
Rutin	−10.0	−58.95	6	THR190 (1.83), HIS41 (2.08), GLY143 (2.39,1.89), ASN142 (1.92), LEU141 (2.10)	-
Myricitrin	−9.1	−49.22	3	CYS145 (2.51), ASN142 (2.04), THR190 (1.83)	π-π stacking HIS41 (5.37)

**Table 4 ijms-22-09431-t004:** The simulation properties of the complexes.

Properties	A	B	C	D	E
RMSD	1.98 ± 0.19	2.25 ± 0.26	3.05 ± 0.57	1.81 ± 0.30	2.26 ± 0.51
RMSF	1.00 ± 0.51	1.31 ± 0.55	1.27 ± 0.58	1.15 ± 0.58	1.25 ± 0.65
rGyr	5.03 ± 0.09	4.96 ± 0.03	4.57 ± 0.05	5.15 ± 0.10	6.09 ± 0.09

Values represent the mean ± SD of 1000 frame index replicates. Letter codes: Quercetin-3-O-Neohesperidoside (A), Quercetin 3-Rhamnoside (B), Myricitrin (C), Rutin (D), and Myricetin 3-Rutinoside (E). Values are in Å.

## Data Availability

Not applicable.

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
