# Peer review of "Development of Effective Therapeutic Molecule from Natural Sources against Coronavirus Protease"

_ijms, 2021, doi:10.3390/ijms22179431_

Round 1

Reviewer 1 Report

This work deals with the in silico evaluation of some natural compounds as inhibitors of Mpro, a COVID-19 relevant protein. The work could be of interest for the readership of the Journal but it should be carefully revised before publication to address the following major points.

1-First, English language level should be improved in all parts of the manuscript.

2-I do not agree with the statement in the Abstract 'To date, neither vaccines nor approved therapeutic agents for effective treatment are available.' There are some vaccines approved and also remdesivir was authorized by FDA as a anti-COVID-19 drug treatment. However, vaccines could probably be a ong-term solution to the problem as vaccination is still slow in several countries and mutations of SARS-CoV-2 could render less effective these prophylactic means, moreover, remdesivir seems to work only with some types of COVID-19 patients. Hence, the importance to continue to investigate new treatments for this disease also using natural products as short-term solution to the COVID-19 problem. Please rewrite that sentence in the Abstract.

3-In the introduction the authors should mention that nucleopeptides and Opuntia-derived phytochemicals were suggested as Mpro potential inhibitors. They should cite at least the works with DOI: 

  • 10.2174/0929867328666210201152326 and 10.3390/sym13061041 

4-the work cited by authors with reference 51 reports a significantly different docking binding energy (about -8 vs. -16 kcal/mol) for Quercetin-3-O-Neohesperidoside docked to Mpro (PDB ID: 6LU7). Please comment, compare, describe in the revised manuscript the differences and the novelty of your findings with respect to those in Ref 51.

5-Quercetin-3-O-Neohesperidoside should be investigated in silico for its druglikeness using one of the several softwares that allow one to assess the druggability of a given compound. 

6-100 nsecs should be 100 ns in any part of the manuscript.

7-please use only one form: MM/GBSA or MM-GBSA

8-Astaxanthin should ne not in capital letters

9-line 91: china should be China

10-line 111: 'This approach is a perfect alternative to experimental techniques.' I do not agree. Docking is complementary to experimental approaches, or can be an alternative but not the perfect one.

11-Figure 1: improve quality of this figure. Moreover, check the structures represented as I saw some exocyclic C=C that I am not sure are expected for some of the phytochemicals.

Author Response

The authors will specifically like to take this time out to appreciate the great job and time invested to review this manuscript. We are indeed grateful for all the constructive concerns raised. We (the authors) have however prioritized all these concerns and further supply adequate responses as requested and for your perusal. Thank you again.

Reviewer 1

Comments and Suggestions for Authors

1-First, English language level should be improved in all parts of the manuscript.

Response:

We have read and corrected all the spelling errors, and also improved the English of this manuscript from the beginning and throughout.

2-I do not agree with the statement in the Abstract 'To date, neither vaccines nor approved therapeutic agents for effective treatment are available. 'There are some vaccines approved and also remdesivir was authorized by FDA as a anti-COVID-19 drug treatment. However, vaccines could probably be a long-term solution to the problem as vaccination is still slow in several countries and mutations of SARS-CoV-2 could render less effective these prophylactic means, moreover, remdesivir seems to work only with some types of COVID-19 patients. Hence, the importance to continue to investigate new treatments for this disease also using natural products as short-term solution to the COVID-19 problem. Please rewrite that sentence in the Abstract.

Response 2: We have completely rewritten the sentence as instructed.

3-In the introduction the authors should mention that nucleopeptides and Opuntia-derived phytochemicals were suggested as Mpro potential inhibitors. They should cite at least the works with DOI: 10.2174/0929867328666210201152326 and 10.3390/sym13061041 

Response 3:

Thank you for this suggestion to back up our argument. We have included them in the introductory section as suggested.

4-the work cited by authors with reference 51 reports a significantly different docking binding energy (about -8 vs. -16 kcal/mol) for Quercetin-3-O-Neohesperidoside docked to Mpro (PDB ID: 6LU7). Please comment, compare, describe in the revised manuscript the differences and the novelty of your findings with respect to those in Ref 51.

Response 4:

This is a good query and the authors really appreciated it. We have been able to compare the energies and suggest possible reasons for the discrepancy. A similar study reported lower negative energy (-8 kcal/mol) for Quercetin-3-O-Neohesperidoside as opposed -16.8 Kcal/mol obtained in this study [46]. The reason could be attributed to the method of preparation and minimization of both the ligand and the receptor and also the size of the generated grid box before docking. In actual fact, the bigger the grid box created prior to docking, the more time consuming the simulation process which could in turn affect the output in terms of binding score.

Although, a number of studies have employed in silico approach to study the interaction of receptor and ligands and also have employed similar compounds for different targets, none of these studies have been able to elucidate or propose the inhibitory mechanism by which flavonoids and their derivatives interact with their respective targets

5-Quercetin-3-O-Neohesperidoside should be investigated in silico for its drug likeness using one of the several softwares that allow one to assess the drugability of a given compound. 

Response 5:

The drug-likeness of all the compounds have now been evaluated using SwissADME and QikProp from Schrodinger as the pharmacological properties while the online software (admetSAR) was used to investigate the toxicity of the compounds.

6-100 nsecs should be 100 ns in any part of the manuscript.

Response 6:

This has been corrected as requested.

7-please use only one form: MM/GBSA or MM-GBSA

Response 7:

We have searched from the beginning and throughout the manuscript and have changed all MM-GBSA to MM/GBSA for uniformity

8-Astaxanthin should ne not in capital letters

Response 8:

This has been corrected as requested.

9-line 91: china should be China

Response 9:

This has been corrected as requested.

10-line 111: 'This approach is a perfect alternative to experimental techniques.' I do not agree. Docking is complementary to experimental approaches, or can be an alternative but not the perfect one.

Response 10:

This has been corrected as requested.

11-Figure 1: improve quality of this figure. Moreover, check the structures represented as I saw some exocyclic C=C that I am not sure are expected for some of the phytochemicals.

Response 11:

Thank you so much for this observation, the structures have been amended for your perusal

Reviewer 2 Report

Comments :

  1. Five bioactive glycosylated flavonoids were identified based on literature search and previously reported by Cherrak et al., [51] as potential Mpro inhibitors. Clerrak et al., also reported already and they have also performed the molecular docking studies. Does this study add any significance/novelty here? Provide details discussion

Reference:  Cherrak, S.A.; Merzouk, H.; Mokhtari-Soulimane, N. Potential bioactive glycosylated flavonoids as SARS-CoV-2 main protease inhibitors: A molecular docking and simulation studies. PloS one 2020, 15, e0240653, doi:10.1371/journal.pone.0240653

  1. Need to check the citation: ex., Line no  “257” : Reference 51 is mentioned as  “Merzouk et al., [51] “ whereas in “287” it is mentioned as. Cherrak et al.,
  2. Line no 249 : The obtained RMSD values range from 1.98±0.19 to 3.05±0.57 Å. Is it for individual compound or whole 5, if it is for the whole it means nothing here provide the same for individual compounds and discuss differences and rank the compounds accordingly?
  3. Comment no 3: same for RMSF, discusses individually
  4. RMSF plot : how does the fluctuation in the active site region? , focus on active site region and discuss along with h-bond interaction.
  5. Line 325 : Refer some more article and ensure the novelty in your claim. Reference 51

Author Response

The authors will specifically like to take this time out to appreciate the great job and time invested to review this manuscript. We are indeed grateful for all the constructive concerns raised. We (the authors) have however prioritized all these concerns and further supply adequate responses as requested and for your perusal. Thank you again.

Reviewer 2

Comments :

  1. Five bioactive glycosylated flavonoids were identified based on literature search and previously reported by Cherrak et al., [51] as potential Mpro inhibitors. Clerrak et al., also reported already and they have also performed the molecular docking studies. Does this study add any significance/novelty here? Provide details discussion

Reference:  Cherrak, S.A.; Merzouk, H.; Mokhtari-Soulimane, N. Potential bioactive glycosylated flavonoids as SARS-CoV-2 main protease inhibitors: A molecular docking and simulation studies. PloS one 2020, 15, e0240653, doi:10.1371/journal.pone.0240653

Response 1

Although studies have used similar approaches to identify the binding interaction between these molecules and different targets, this study was specifically designed to uncover the mechanism of interaction of flavonoids and their derivatives with targets. We have explicitly discussed this difference with Quercetin-3-O-Neohesperidoside specifically in figure 3. In addition, our study identified Quercetin-3-O-Neohesperidoside as better inhibitory molecule against SARS-CoV-2 Mpro among other studied molecules

  1. Need to check the citation: ex., Line no  “257” : Reference 51 is mentioned as  “Merzouk et al., [51] “ whereas in “287” it is mentioned as. Cherrak et al.,

Response 2:

We have corrected this to Cherrak et al. (The first author’s name)

  1. Line no 249 : The obtained RMSD values range from 1.98±0.19 to 3.05±0.57 Å. Is it for individual compound or whole 5, if it is for the whole it means nothing here provide the same for individual compounds and discuss differences and rank the compounds accordingly?

Response 3:

Rutin has the lowest RMSD (1.81±0.30 Å) followed by Quercetin-3-O-Neohesperidoside with RMSD of 1.98±0.19 Å. The lower the RMSD the less fluctuation and the greater the stability [54]. The stability of these compounds are in the order of Rutin > Quercetin-3-O-Neohesperidoside > Quercetin 3-Rhamnoside > Myricetin 3-Rutinoside > Myricitrin.

  1. Comment no 3: same for RMSF, discusses individually

Response 4:

The RMSFs of the complexes were ≤ 1.31 Å. Quercetin-3-O-Neohesperidoside has the lowest RMSF of 1.00±0.51 Å followed by Rutin with RMSF of 1.15±0.58 Å. Quercetin 3-Rhamnoside, Myricitrin, and Myricetin 3-Rutinoside have the highest RMSF of 1.31±0.55, 1.27±0.58, and 1.25±0.65 Å, respectively.

  1. RMSF plot : how does the fluctuation in the active site region? , focus on active site region and discuss along with h-bond interaction.

Response 5:

The RMSF values of the systems relative to each amino acid residue of the complexes were evaluated to compare the residues’ flexibility. The RMSF reported the flexible region of the complexes and displayed that none of the important catalytic residues present in the active site region has shown an RMSF value of more than 1.31 Å. This serves to confirm that the catalytic machinery present in the Mpro active site was not distorted upon binding of the compounds. The amino acids, namely, THR26, HIS41, CYS44, SER47, MET49, TYR54, ASN142, GLY143, SER144, CYS145, HIS164, MET165, GLU166, LEU167, ASP187, THR190, GLN192, ARG217, THR304, and PHE305 have displayed flexibility upon the binding of the studied compounds. Regardless of the nature of these residues within the active site of Mpro, they were relatively stable in complex systems indicating the binding nature of the compounds. With respect to all these complexes, the residues fluctuate with respect to the nature of the compound in the active site and types of interacting atoms. Particularly, THR190, GLU16, HIS163, PHE140, GLY143, CYS145, and GLN189 fluctuated more compared to other amino acids in the active site indicating their flexible nature and therefore could be a good determinant in the identification of inhibitory molecules.

  1. Line 325 : Refer some more article and ensure the novelty in your claim. Reference 51

Response 6:

We have added the novelty of the claim in the body of the discussion. The stated line 325 from the initial upload is part of the conclusion which stated ‘Transmission’.

Round 2

Reviewer 1 Report

The manuscript can be published in the current form 

Reviewer 2 Report

author has revised the manuscript and may be proceed further